# Polymer Dispersed Cholesteric Liquid Crystals with a Toroidal Director Configuration under an Electric Field

**DOI:** 10.3390/polym13050732

**Published:** 2021-02-27

**Authors:** Anna P. Gardymova, Mikhail N. Krakhalev, Victor Ya. Zyryanov, Alexandra A. Gruzdenko, Andrey A. Alekseev, Vladimir Yu. Rudyak

**Affiliations:** 1Institute of Engineering Physics and Radio Electronics, Siberian Federal University, 660041 Krasnoyarsk, Russia; gard@iph.krasn.ru (A.P.G.); kmn@iph.krasn.ru (M.N.K.); 2Kirensky Institute of Physics, Federal Research Center KSC SB RAS, 660036 Krasnoyarsk, Russia; zyr@iph.krasn.ru; 3Faculty of Physics, Lomonosov Moscow State University, 119991 Moscow, Russia; gruzdenko@polly.phys.msu.ru (A.A.G.); alekseev@polly.phys.msu.ru (A.A.A.)

**Keywords:** cholesteric droplet, orientational structure, electric field, polymer dispersted liquid crystal, dichroic dye

## Abstract

The electro-optical properties of polymer dispersed liquid crystal (PDLC) films are highly dependent on the features of the contained liquid crystal (LC) droplets. Cholesteric LC droplets with homeotropic boundaries can form several topologically different orientational structures, including ones with single and more point defects, layer-like, and axisymmetric twisted toroidal structures. These structures are very sensitive to an applied electric field. In this work, we have demonstrated experimentally and by computer simulations that twisted toroidal droplets reveal strong structural response to the electric field. In turn, this leads to vivid changes in the optical texture in crossed polarizers. The response of droplets of different sizes were found to be equivalent in terms of dimensionless parameters. In addition, the explanation of this phenomenon showed a comparison of theoretical and experimental structural response curves aids to determine the shape of the droplet. Finally, we demonstrated that the addition of a dichroic dye allows such films to be used as optical filters with adjustable color even without polarizers.

## 1. Introduction

Polymer dispersed liquid crystal (PDLC) films combine the useful properties of liquid crystals (LC) and polymers. As a result, PDLC films have unique optical and dielectric properties and allow to control it by an electric field [1,2,3], temperature [4,5], light radiation [6,7], geometrical confinement, and mechanical strain [8,9]. On the other hand, PDLCs are characterized by simplicity and manufacturability of production, usability, and resistance to environmental influences [10]. The properties of PDLC films and their change under external stimuli depend on the director **n** configuration formed in the LC cavities. In turn, the director orientational structure depends on the LC cavity size, boundary conditions, and material parameters of the LC [11]. A nematic LC in spheroidal cavities (LC droplets) with tangential boundary conditions forms a bipolar/twisted-bipolar structure with a pair of point surface defects (boojums) [12,13] or a toroidal configuration with a linear defect in the bulk [14]. Homeotropic boundary conditions lead to the formation of a radial structure with a bulk point defect in the center of the droplet (a `hedgehog’) or an axial configuration with a circular surface defect at the droplet equator [15]. An axial-bipolar configuration with a pair of boojums and a circular surface defect [13,16,17] or a twisted structure with a pair of boojums at the poles and a `hedgehog’ in the center of the droplet is formed in nematic under conical boundary conditions [18]. The application of an electric or magnetic field changes the orientation of the bipolar axis of the droplets in the case of tangential or conical boundary conditions [15,19] or switches the director structure from radial to axial in the case of homeotropic anchoring [20]. This leads to a change in light scattering [21,22,23] and absorption [24,25] of PDLC film, which can be used in optical shutters and smart films.

The addition of a chiral agent to the nematic in a free state leads to the formation of a helicoid structure of the director **n** field (cholesteric liquid crystal, CLC). Consequently, plenty of various director configurations emerge in CLC droplets. In this case, an additional parameter determining the director orientational structure is the relative chiral parameter $N_{0}$, which equals to twice the ratio of the droplet diameter *d* to the intrinsic helix pitch $p_{0}$ ($N_{0}=2d/p_{0}$). This ratio shows the number of director π turns along the droplet diameter *d* in an equilibrium free standing cholesteric. The $N_{0}$ value determines the type of orientational structure in the CLC droplet. Under tangential boundary conditions, the twisted bipolar structure (at $N_{0}<2$) [26], an intermediate configuration for which the distance between boojums decrease with an increase to the $N_{0}$ value (at $2\leq N_{0}\leq 5$) [27], and a layer-like structures with a radial or a diametrical linear defect (at $N_{0}>5$) [28,29] were obtained. A twisted axial-bipolar structure (at $N_{0}<2.9$) with a circular surface defect at the droplet equator, an intermediate configuration with a twisted circular defect (at $2.9\leq N_{0}\leq 3.95$), and a layer-like structure with a double twisted defect loop (at $N_{0}>3.95$) are formed in cholesteric droplets under conical boundary conditions [30,31]. Homeotropic boundary conditions in cholesteric droplets frustrate the helicoidal structure. As a result, structures with different numbers of point defects in the bulk and/or near the surface (at $N_{0}<5.5$), layer-like structures (at $N_{0}>2.5$), and an axisymmetric twisted toroidal structure (at $2.9<N_{0}<5.4$) were obtained [32,33,34,35].

The formation of complex cholesteric structures expands the scope of applications of PDLC films, for instance, in electrically controlled reflectors [36,37], color electronic paper [38], printable temperature sensors [39], systems for lasing [40,41,42], solar cells [43], etc. On the other hand, the structure control inherent in flat CLC layers comes with its own difficulties. These are complex dynamics of both the response to external actions and subsequent relaxation, higher values of the control electric/magnetic fields, bi- and multi-stability, etc. The electric field unwinds the CLC helix and transforms the cholesteric into a nematic state in a sufficiently strong field in the case of a positive anisotropy of permittivity Δε [44]. Similar unwinding induced by surface anchoring is observed in cholesteric droplets with homeotropic boundary conditions. In particular, this led to a decrease in the observed number of π turns of the director along the droplet diameter *d* in comparison with $N_{0}$ [45]. A similar effect was observed for droplets with an axisymmetric twisted toroidal configuration, in which the maximum turn angle of the director along the droplet diameter equals to π regardless of the value $2.9<N_{0}<5.4$ [35]. Since the optical response to an applied electric field is a practical matter for this system, important questions arise: How does it depend on the magnitude of the field, and will the characteristic ratio $N_{0}$ alter the response or not?

In this paper, we study PDLC films with embedded cholesteric LC droplets in an axisymmetric twisted toroidal configuration to answer the question. We examine the action of an electric field perpendicular to the film plane by both experiment and computer simulations. Comparing the results of both methods, we make conclusions about the origins of the observed electro-optical response of the films, the dependence of the droplet shape on its size, and the detailed structures of cholesteric droplets. While the main results suppose the use of crossed polarizers, we demonstrate the possibility of using PDLC films without polarizers as an optical filter with adjustable color saturation by adding a dichroic dye to the CLC mixture.

## 2. Materials and Methods

### 2.1. Experimental Approach

The polymer dispersed liquid crystals (PDLC) films based on the nematic mixture E7 (Merck) doped with the left-handed chiral dopant cholesteryl acetate (Sigma Aldrich) were studied. The cholesteryl acetate concentrations were 1 wt % and 4 wt % correspondings to the intrinsic helix pitch $p_{0}=14\;\mu$m and $p_{0}=4\;\mu$m. The 4 μm helix pitch CLC was doped with a dichroic dye Blue AB4 (Nematel) with weak absorption in the blue spectrum region and the dichroic ratio was 13.6 at 641 nm. The dichroic dye concentration was 2 wt %. Liquid crystal was dispersed in poly(isobutyl methacrylate) (Sigma Aldrich, St. Louis, MO, United States), which specifies homeotropic boundary conditions for E7 nematic. The sandwich-like cells were manufactured by combining technology: The solvent-induced phase separation (SIPS) and thermal-induced phase separation (TIPS) [10]. The glass substrates coated with ITO have been used. At the first stage (SIPS), CLC was added to the 10% solution of PiBMA in butyl acetate. The weight ratio of cholesteric and polymer components was CLC:PiBMA = 60:40. The prepared homogeneous solution was poured on the glass substrate and dried for 24 h. At the second stage (TIPS), PDLC film was covered with the second substrate and heated to 80 ${}^{{}^{\circ}}$C under pressure. Then it was cooled down to room temperature for 5 h, resulting in the PDLC cell under study. PDLC film thickness *H* was specified by a 20 μm diameter glass microspheres (Duke Scientific Corporation, Palo Alto, CA, United States), which were added to the solution of PiBMA in butyl acetate at the first stage (SIPS). The droplet sizes *d* (apparent diameter of the droplet) were determined by the sample cooling rate and were found to be in the range from 6 to 35 μm for a 14 μm helix pitch CLC and in the range from 3 to 15 μm for a 4 μm helix pitch CLC. Experimental studies were carried out using a polarizing optical microscope (POM) Axio Imager.A1m (Carl Zeiss) at $t=22{\;}^{{}^{\circ}}$C. An AC electric field (1 kHz) from a low-frequency generator of electrical signals G3-123 was applied perpendicular to the PDLC film.

### 2.2. Computer Simulations

#### 2.2.1. Calculations of the Droplet Structure

We performed calculations of the LC structure within oblate ellipsoidal droplets with homeotropic boundaries filled with chiral nematic. The volume was rendered in lattices from 48×48×24 to 48×48×48 points depending on the height of the droplet *h*. The other two dimensions were equal to *d* (droplet diameter), and the compression ratio δ=h/d varied from 0.5 (oblate ellipsoid) to 1 (sphere). We used extended the Frank elastic continuum approach with Monte-Carlo annealing optimization [46] to find the energy-optimal droplet structures. This approach includes the effects of the director field distortion and the formation of defects in the droplet:(1)$$\begin{array}{c} F=\int_{V}\left(\frac{K_{11}}{2}{\left(\mathrm{div}n\right)}^{2}+\frac{K_{22}}{2}{\left(n\cdot \mathrm{rot}n+q_{0}\right)}^{2}+\frac{K_{33}}{2}{\left[n\times \mathrm{rot}n\right]}^{2}\right)dV+\frac{W}{2}\int_{\Omega }\left(1-\cos^{2}\gamma \right)d\Omega +F_{def}+\varepsilon_{0}\Delta \varepsilon \int_{V}\left(E^{2}-{\left(E\cdot n\right)}^{2}\right)dV, \end{array}$$
where $K_{11}$, $K_{22}$, and $K_{33}$ are the splay, twist, and bend elasticity constants, respectively, $q_{0}=2\pi /p_{0}$, *W* is the surface anchoring energy density, γ is the angle between local director and normal to the droplet surface, $F_{def}$ is the energy of defects calculated by the summation of the point and linear defect energies, and Δε is dielectric anisotropy of LC, and E is the electric field applied along the short axis (*z*). The types, positions, and energies of defects were estimated automatically during the Monte-Carlo optimization procedure (see details in [46]). The ratio between elasticity constants was set to $K_{11}:K_{22}:K_{33}=1:0.685:1.54$ to simulate the cholesteric liquid crystal mixture under study. To take into account the potential formation of the disclination lines with core, its linear energy density was set to $f_{core}^{line}=2.75K_{11}$ (same as in [35]). The relative chiral parameter $N_{0}$ was varied from 2.7 to 4.3. The homeotropic anchoring strength was set to strong, $\mu =Wd/2K_{11}$ varied from 1500 (for $N_{0}=2.7$) to 2400 (for $N_{0}=4.3$) to maintain constant natural units value of $W=1.75\times 10^{-3}$ J/m${}^{2}$. To simulate changes in the structure under a slowly increasing electric field, we increased the the amplitude of the electric field in small steps and applied Monte-Carlo relaxation at each step. The data are shown in dimensionless electric field, which were calculated as $e=|E|d{\left(\varepsilon_{0}\Delta \varepsilon /4K_{11}\right)}^{1/2}$.

#### 2.2.2. Calculation of the POM Textures

We calculated the POM textures of spherical-cap droplets using the Jones matrix technique, formulated for PDLC materials in [47]. This technique supposes direct unidirectional propagation of linearly polarized light through a non-uniform birefringent material. Light diffraction, diffusion, and scattering are not taken into account in the Jones calculus, and thus textures on the peripheral parts of the droplets are roughly estimated. Textures were calculated for 10 different wavelengths within the visible spectrum, from 400 nm to 700 nm with an equal step of 33 nm. The values of ordinary and extraordinary reflective indices for E7 cholesteric were set in according to [48]. Color textures were created by merging the individual wavelength textures with regard to the luminescence spectra of the black body at temperature ≈3000 K.

## 3. Results and Discussion

In the experiment, we obtained radial, twisted radial, twisted toroidal, and layered structures. In accordance to the previous study [35], multiple types of structures can be formed in droplets with the same chiral parameter $N_{0}$. A twisted toroidal configuration was observed in the range of $2.9\leq N_{0}\leq 5.8$, while at $N_{0}<2.9$ only structures with one point defect were formed, and at $N_{0}>5.8$ only layer-like configurations were formed. Computer simulations have shown that the twisted toroidal configuration has the lowest energy at $3.2<N_{0}<5.3$. For smaller droplets, the toroidal configuration is metastable. At the same time, it is stable enough up to $N_{0}=2.7$ to behave similarly under relaxation under the action of an electric field, which additionally stabilizes this director configuration.

### 3.1. Effect of an Electric Field on the Twisted Toroidal Structure

The twisted toroidal structure has an axial symmetry of the director field with a near-surface circular defect (Figure 1a). The calculated orientational structure is shown in Figure 1f. Initially, two stable orientations of the structure symmetry axis were observed in the experimental samples: Perpendicular and parallel to the PDLC film plane. The absence of intermediate orientations of the symmetry axis can be explained by the oblate ellipsoidal shape of the droplets with a short axis oriented perpendicular to the film plane [10]. Accordingly, the most stable states correspond to the structure symmetry axis orientations parallel to the ellipsoid primary axes.

When the symmetry axis of the toroidal structure is oriented perpendicular to the electric field, the reorientation process in a slowly increasing electric field can be split into three stages. Figure 1a–e demonstrates the optical microscopy observations of a CLC droplet with $N_{0}=3.9$. First, below E≲0.11 V/μm this structure stays almost unchanged, with only minor changes in the central part of the droplet. Then, in a narrow range from 0.11 V/μm to 0.13 V/μm, the structure symmetry is broken, the defect ring distorts and begins to rotate (Figure 1b,c). At this stage, switching off the field will return the structure to the initial state. Finally, at E≳ 0.13 V/μm, the distorted structure becomes unstable, and the entire structure transforms into an axisymmetric twisted toroidal state with a symmetry axis along the electric field, as well as a short axis of the droplet (Figure 1d). The new orientation of the toroidal structure remains stable after switching off the field (Figure 1e). Thus, a single application of a sufficiently strong electric field aligns all CLC droplets with a twisted toroidal structure, and circular defects remains parallel to the film plane after the electric field is turned off.

One of the essential features of the toroidal configuration is the central area of the droplets, where the director is oriented almost parallel to the symmetry axis. In POM this area appears as a dark zone surrounded by a bright circle or parts of a circle (Figure 2a). The applied electric field orients the director perpendicular to the sample plane and expand the dark area. The experimentally measured ratio of the dark area diameter to the droplet size a/d has similar values for droplets of different size in the absence of an electric field (Figure 2b). Under an electric field, measurements show S-shaped curves for the a/d(E) dependencies: Under a small field, the dark area remains almost the same, then it grows rapidly to a/d∼0.6 in the medium fields, then at higher *E* it slowly approaches to 0.9. Small droplets require higher values of control voltage to achieve the same a/d values (see Appendix A). Surprisingly, the dependencies of the ratio a/d(e) on the dimensionless electric field $e=Ed{\left(\varepsilon_{0}\Delta \varepsilon /4K_{11}\right)}^{1/2}$ almost coincide for droplets with different $N_{0}$ values (Figure 2b).

In computer simulations, the initial toroidal structures in the absence of an electric field strongly depend on both $N_{0}$ and the droplet compression ratio δ. In the largest droplets considered ($N_{0}=4.3$), it varies from 0.22 for spherical droplet to ≈0.6 for δ=0.50 (see Figure 3). The orientational structure changes under an electric field vary correspondingly. For the near-spherical droplets, the changes in POM images and a/d(e) curves are very similar to the experimental data. However, highly oblate droplets showed a significantly faster growth of the dark zone under medium electric fields up to a/d∼0.8, which is close to the highest obtained value of a/d (for example, see δ=0.50 curve in Figure 3). In all cases, the circular defects show only minor growth under an electric field. Thus, the most changes in the director orientation occur in the zone between the defect and dark zone, where the director undergoes high elastic deformations resulting into bright tangential POM lines. This area diminishes under an electric field, resulting into the increase in the diameter of the dark zone *a*. In highly oblate droplets, the size of this area is small even in the absence of an electric field; therefore, there is a large dark zone and weak or no response to the electric field (see data for all $N_{0}$ values in Appendix A). At the same time, the maximum total turn of the director along the droplet diameter remains equal to π in all cases.

To demonstrate the primary changes in response curves a/d(e) over droplets of different sizes and compression ratios, we plotted in Figure 4
a/d dependencies on δ and $N_{0}$ for the cases of e=0 and e=4.2 (the steepest part of the response curve for most of the systems). Ratio a/d increases with increasing droplet compression at all values of *e*. At the same time, larger droplets show a smaller characteristic ratio a/d. Notably, the comparison of these data with experimental results allows one to roughly estimate the compression ratios of the droplets in the real material in an independent and non-destructive way.

Based on the data of experiment and computer simulations, we came to the conclusion that the larger the experimentally observed droplet, the smaller the compression ratio δ it has [49]. Figure 5 shows the suggested choice of the dependency $\delta (N_{0})$ and the resulting set of curves a/d(e). It allows one to explain the similar values of a/d for different droplet sizes in the absence of an electric field as well as the coincidence of the experimentally measured a/d curves. At the same time, it should be mentioned that the experimental data and the results of computer simulations does not completely coincide. First, the initial size of the dark zone a/d(e=0) is higher in the simulations (0.32–0.34 instead of 0.14–0.19 in the experiment). Second, the response to the electric field in the simulations occurs at ≈1.5 times lower values of *e*. We presume that both issues originate from the fact that the simulated POM images do not take into account the lens effect of droplets due to the difference in refractive indices between the polymer and LC.

It should be noted that the system under study demonstrates no hysteresis in the electrically induced response of the twisted toroidal structure. The dependencies of ratio a/d(E) at increasing and decreasing AC electric field are the same. Thus, the application of an electric field allows to control the *a* size of the central droplet region, where the director is oriented almost parallel to the field, while the director still turns $90^{{}^{\circ}}$ along the rest of the distance (d−a)/2.

### 3.2. The Response of Cholesteric Droplets Doped with a Dichroic Dye

The character of the response of the orientational structure to an electric field in droplets with a twisted toroidal configuration allows for the smooth and precise control of the optical parameters of CLC droplets by an electric field. In some cases, this is even possible without polarizers. For instance, the color of a cholesteric LC doped with a dichroic dye depends on the director orientation with respect to the observation direction. As a consequence, the central part of a droplet with a twisted toroidal configuration will be practically colorless (Figure 6a). Apparently, it is the same zone that is dark under crossed polarizers. Thus, the diameter a(E) of the colorless central area of the droplet can be similarly controlled by the electric field (Figure 6b–d). Simultaneously, the color saturation of the droplet peripheral area practically does not change under the action of the electric field since the total maximum director turn angle is preserved. Each CLC droplet in the PDLC film can be considered as an electrically controlled diaphragm with tunable color saturation. Thus, the effective diaphragm size will depend on the applied voltage similarly to the dependence of the a/d ratio on the dimensionless electric field *e*, which allows to control the color saturation of the film by the electric field.

## 4. Conclusions

In this work, we studied by means of experiment and computer simulations the electro-optical response of the toroidal orientational structure in spheroid cholesteric droplets with homeotropic boundary conditions to the application of an electric field. In the absence of an electric field, a small dark zone occurred in the central part of the pre-aligned droplets in crossed polarizers microscopy. The size of the dark zone increased when a moderate electric field was applied perpendicular to the PDLC film. At 1–1.5 V/μm, the diameter of the dark zone exceeded 0.8 of the droplet diameter (the exact voltage depends on the cholesteric pitch to the droplet size ratio). Computer simulations confirmed this behavior and suggest a direct interrelation between droplet size and its compression ratio (the larger the droplet, the flatter it is), which was proven previously [49]. The structural changes in the toroid configuration were found to be fully reversible in a broad range of applied voltage. It allows one to adjust the optical parameters of both the LC droplet and, consequently, the whole PDLC film in a smooth and controllable manner. We demonstrated that the addition of a dichroic dye allows one to obtain an optical response even without polarizers. In this case, cholesteric liquid crystal droplets act as a diaphragms with adjustable color saturation, gradually changing the naked eye-observable saturation and brightness of the material under the action of an electric field. We believe this may significantly advance optical materials for many applications (such as printable sensors and electrically-controlled color filters) due to multiple advantages, including preparation simplicity and scalability for these materials, their thermal and mechanical stability, and high potential for further adjustment for application requirements.

## Figures and Tables

**Figure 1 polymers-13-00732-f001:**
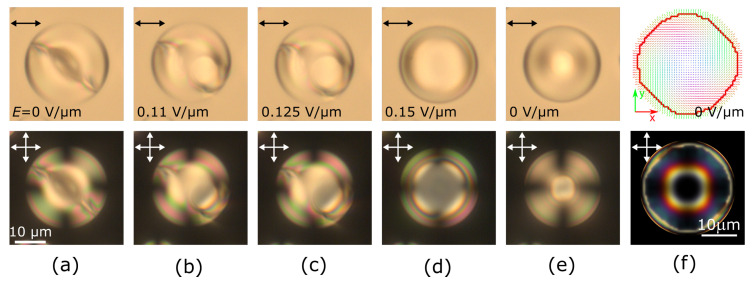
Experimental polarized optical microscopy images (**a**–**e**) of the cholesteric droplet at $N_{0}=3.9$ taken with one polarizer (top row) and in the crossed polarizers (bottom row). Initial state with the circular defect perpendicular to the film plane (**a**), under an electric field *E* equal to 0.11 V/μm (**b**), 0.125 V/μm (**c**), 0.15 V/μm (**d**), and after switching off the voltage (**e**). Intrinsic helix pitch is $p_{0}=14\;\mu$m, the scale remains the same on the images (**a**–**e**). Calculated director distribution (**f**, top row) and the corresponding simulated polarized optical microscopy image (**f**, bottom row). Hereinafter, double arrows indicate the direction of the polarizers, and the director n is colored in correspondence with the direction (red along the *x*-axis, green along the *y*-axis, blue along the *z*-axis), the thick red line indicates the circular defect near the surface.

**Figure 2 polymers-13-00732-f002:**
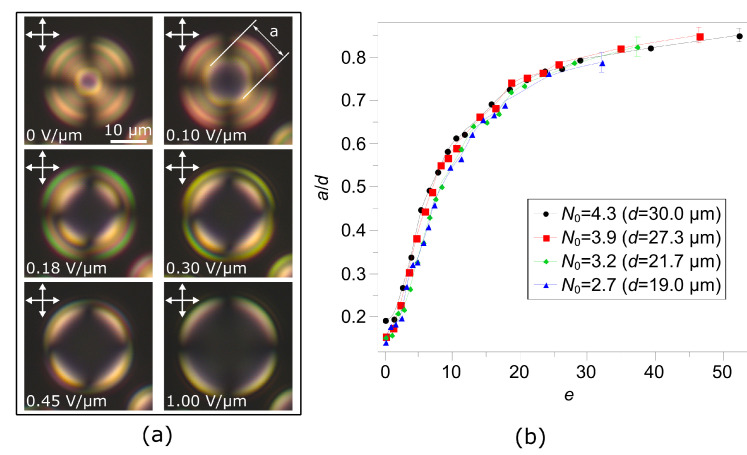
Polarizing optical microscope (POM) images of the cholesteric droplet at $N_{0}=3.9$ taken in the crossed polarisers for different applied electric fields values; the scale remains the same in all images (**a**). Dependencies of the a/d ratio on the applied dimensionless electric field *e* obtained for CLC droplets at different values of $N_{0}$ (**b**). Intrinsic helix pitch is $p_{0}=14\;\mu$m.

**Figure 3 polymers-13-00732-f003:**
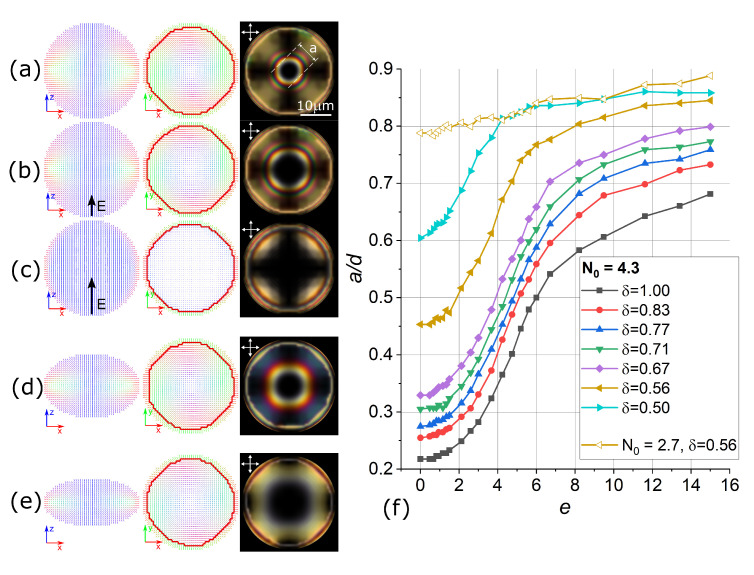
Calculated director distributions in the central cross-sections and simulated POM images for $N_{0}=4.3$ in spherical droplet at e=0 (**a**), e=4.2 (**b**), e=15 (**c**), and in oblate droplets with δ=0.67 (**d**), and δ=0.5 (**e**) at e=0. Calculated dependency a/d(e) for $N_{0}=4.3$ in droplets of various shapes (**f**). The scale remains the same in all images.

**Figure 4 polymers-13-00732-f004:**
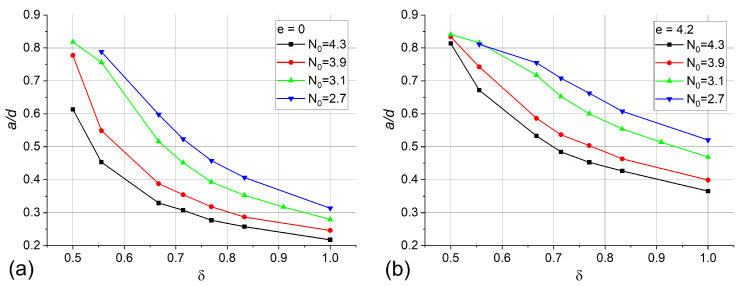
Calculated dependencies of a/d ratio on droplet compression ratio δ at no electric field (**a**) and at e=4.2 (**b**).

**Figure 5 polymers-13-00732-f005:**
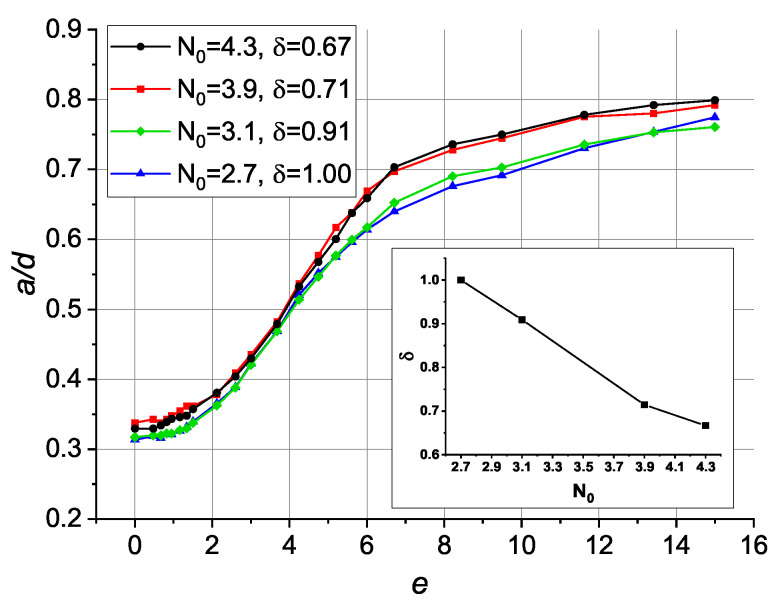
Calculated dependencies of relative dark zone size a/d on the dimensionless electric field *e* for droplets with various chiral parameter $N_{0}$ and arbitrary compression ratio δ. The inset shows $\delta (N_{0})$ dependency.

**Figure 6 polymers-13-00732-f006:**
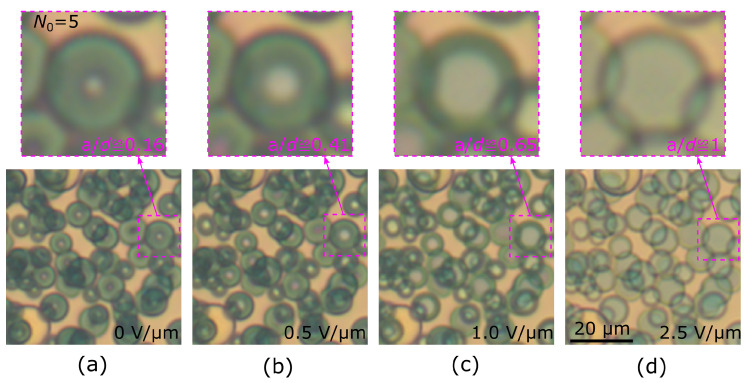
Optical images of the cholesteric droplets taken in non-polarized light. Electric field *E* equals to 0 V/μm (**a**), 0.5 V/μm (**b**), 1.0 V/μm (**c**), and 2.5 V/μm (**d**). The insets show the droplet with $N_{0}=5$. Intrinsic helix pitch is $p_{0}=4\;\mu$m. The scale remains the same in all images in a row.

## Data Availability

The data presented in this study are available on request from the corresponding author.

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
