# Peer review of "Polymer Dispersed Cholesteric Liquid Crystals with a Toroidal Director Configuration under an Electric Field"

_polymers, 2021, doi:10.3390/polym13050732_

Round 1

Reviewer 1 Report

The manuscript entitled is a valid research work. The topic of the manuscript fits well the scope of the journal. The topic is very highlighted and will be of high interest to the readers of Polymers. I expect this manuscript will be appropriately cited if published as it will definitely have an appropriate impact on the specific field of research. The results presented in the manuscript are appropriately described and treated; they are novel and original. The technical quality of the work is high. There is no doubts that this manuscript can be considered for acceptance at Polymers. However, there are several issues, mostly technical ones, that requires author's attention. Here they are:

(A) The style of the manuscript language should be carefully checked by the native speaker; this will increase the overall clarity of the work.

(B) The state-of-the-art in the specific field of research is well presented in the introduction section. However, the references supported are not always the latest ones and might be updated. This can increase the added value of the whole work. Specifically, authors can consider the following and recent works to be sited at appropriate places of the manuscript:

(i) on macromolecular PDLC: [Phys. Chem. Chem. Phys., 2020, 22, 23064-23072. DOI: 10.1039/D0CP04143B]

(ii) page 1; line 17: beyond ref 4, please consider to use [Polymer 2018, 158, 96-102 Doi: 10.1016/j.polymer.2018.10.049]

(iii) page 2;line 50: reference of solar cells might be added [Prog. Photovolt: Res. Appl., 2001, 9: 263-271. Doi: 10.1002/pip.378]

(iv) general relevant reference: [J. Phys.: Condens. Matter 29 (2017) 133003 Doi: 10.1088/1361-648X/aa5706]

(v) on optical and electro-optical properties: [Phys. Rev. E 2007,  75, 011705. Doi: 10.1103/PhysRevE.75.011705] and [Crystals 2019, 9, 282; doi:10.3390/cryst9060282] and [Molecular Crystals and Liquid Crystals, 494:1, 242-251, DOI: 10.1080/15421400802430430]

(C) There are quite a lot of the microphotographs in this manuscript. However, the scale bar is not presented on all of them. Authors should carefully check this issue and clarify it. Does it mean that if the scale bar presented on one of the sub-figure, the scale of all the rest sub-figures is the same? See for ex. Figs. 1 & 2. 

(D) On all sub-figures of Figure 3 and 4, the scale bar is missing. Please revise.

Summary: The manuscript can be considered for acceptance after a minor and careful revision as indicated above.

Author Response

REVIEWER 1

Comment:

The manuscript entitled is a valid research work. The topic of the manuscript fits well the scope of the journal. The topic is very highlighted and will be of high interest to the readers of Polymers. I expect this manuscript will be appropriately cited if published as it will definitely have an appropriate impact on the specific field of research. The results presented in the manuscript are appropriately described and treated; they are novel and original. The technical quality of the work is high. There is no doubts that this manuscript can be considered for acceptance at Polymers. However, there are several issues, mostly technical ones, that requires author's attention. Here they are:

Answer:

We would like to thank the Reviewer for the attention to our manuscript and his/her high mark. The issues shown by the Reviewer helped us to improve the manuscript. We have updated it accordingly and believe now it can be accepted to the Polymers journal.

Comment:

(A) The style of the manuscript language should be carefully checked by the native speaker; this will increase the overall clarity of the work.

Answer:

We agree with the Reviewer and apologize for many mistakes in the manuscript. The whole text was checked and corrected.

Comment:

(B) The state-of-the-art in the specific field of research is well presented in the introduction section. However, the references supported are not always the latest ones and might be updated. This can increase the added value of the whole work. Specifically, authors can consider the following and recent works to be sited at appropriate places of the manuscript:

(i) on macromolecular PDLC: [Phys. Chem. Chem. Phys., 2020, 22, 23064-23072. DOI: 10.1039/D0CP04143B]

(ii) page 1; line 17: beyond ref 4, please consider to use [Polymer 2018, 158, 96-102 Doi: 10.1016/j.polymer.2018.10.049]

(iii) page 2;line 50: reference of solar cells might be added [Prog. Photovolt: Res. Appl., 2001, 9: 263-271. Doi: 10.1002/pip.378]

(iv) general relevant reference: [J. Phys.: Condens. Matter 29 (2017) 133003 Doi: 10.1088/1361-648X/aa5706]

(v) on optical and electro-optical properties: [Phys. Rev. E 2007,  75, 011705. Doi: 10.1103/PhysRevE.75.011705] and [Crystals 2019, 9, 282; doi:10.3390/cryst9060282] and [Molecular Crystals and Liquid Crystals, 494:1, 242-251, DOI: 10.1080/15421400802430430]

Answer:

Thank you for your comment. We agree that these references are relevant and useful for the potential reader. We added it to the manuscript.

Comment:

(C) There are quite a lot of the microphotographs in this manuscript. However, the scale bar is not presented on all of them. Authors should carefully check this issue and clarify it. Does it mean that if the scale bar presented on one of the sub-figure, the scale of all the rest sub-figures is the same? See for ex. Figs. 1 & 2.

Answer:

Thank you for this comment. Yes, the single scale bar for sub-images means the same scale for all sub-images. We inserted corresponding comments to the images. 

Comment:

(D) On all sub-figures of Figure 3 and 4, the scale bar is missing. Please revise.

Answer:

Thank you for noting this. We inserted the scale bar to the simulated XPOL images on Fig. 1 and Fig. 3 and 4 (was Fig. 4 before revision). Please note, that Fig. 4 is now Fig. 3, as we moved one from the main text to the Supplementary according to the comments of the Reviewer 2.

Reviewer 2 Report

The manuscript titled “Polymer Dispersed Cholesteric Liquid Crystals With a Toroidal Director Configuration under an Electric Field” describes the behavior of cholesteric droplets dispersed in a polymer matrix under the action of electric field. Under the field applied, LC molecules align along the field vector (vertically to the sample plane) yielding back circular areas observed between cross polarizers. Some dependencies of the optical effect, as well as computer simulation, are presented.  The work looks unfinished with unclear novelty and impact. The choice of the journal is also very questionable, since the paper is not about polymers at all, and the fact that the system is liquid crystals dispersed in polymer matrix doesn’t automatically make it appropriate for “Polymers”. Unfortunately, I cannot support this work for possible publication.

Here are some additional comments which might help the authors to improve their work:

  • In my opinion, the strong point of the work is the tunability of optical properties (specifically, colors) of the PDLC films, as it is highlighted in the abstract, however, the manuscript is mainly about calculations. I would very much recommend the authors reconsider the last section of the article and provide more experimental results proving the “tunability of color”. Now, it can hardly be said from Figure 6 that any color tunability takes place. Additional spectral characterization is also needed.
  • The novelty of the work should be expressed clearly.
  • Figure 3 or Figure 4 seems unnecessary. There is no need to show quite similar results for different N. One of them can be easily shifted to supporting information, and the discussion on simulations could also be sharpened.
  • Few comments regarding the experimental part: 1) It is not indicated how much dichroic dye was used in CLC. 2) How CLC was dispersed in PiBMA? It is not possible to do since the polymer is solid. So, please indicate how it was done (solvent, etc.). 3) How the PDLC films were fabricated?

Author Response

REVIEWER 2

Comment:

The manuscript titled “Polymer Dispersed Cholesteric Liquid Crystals With a Toroidal Director Configuration under an Electric Field” describes the behavior of cholesteric droplets dispersed in a polymer matrix under the action of electric field. Under the field applied, LC molecules align along the field vector (vertically to the sample plane) yielding back circular areas observed between cross polarizers. Some dependencies of the optical effect, as well as computer simulation, are presented.  The work looks unfinished with unclear novelty and impact. The choice of the journal is also very questionable, since the paper is not about polymers at all, and the fact that the system is liquid crystals dispersed in polymer matrix doesn’t automatically make it appropriate for “Polymers”. Unfortunately, I cannot support this work for possible publication.

Answer:

First of all, we would like the Reviewer for the useful comments that helped us to improve the quality of the manuscript. 

Regarding the choice of the journal, we should mention that the manuscript was submitted to the special issue of Polymers journal “Polymer-Dispersed Liquid Crystals, Fundamental Principles, Materials, Technology, and Applications”, which focuses on a variety of topics on the physics and chemistry of relevant materials, technology, and applications of polymer-dispersed liquid crystals. The manuscript addresses the problem of electro-optical properties of PDLC material and its fundamental physical background investigated by the combination of experimental and computer simulation techniques. It should be noted that the properties of PDLC materials are determined by the interplay between polymer and LC materials used. The polymer is important not only as a useful ‘carrier’ of LC droplets, but actually gives rise to a lot of effects and applications. For example, the non-spherical shape of the LC droplets and its dependency on size, the temperature control on the droplets, the boundary conditions on LC-polymer surface and the corresponding orientational structures, the elasto-optical effects are all direct results of interaction between polymer and LC. It’s incorrect to consider these effects as solely ‘liquid crystalline physics’ independent of the polymer part.

The manuscript shows several important features of cholesteric PDLC materials viable for future applications. We demonstrated that (1) with cross-polarizers, it is possible control the light transmittance by electric field in PDLC materials under study; (2) the fundamental origins of the optical changes grant the unified value of reduced transition electric field for the droplets of different sizes and shapes; (3) the same PDLC materials allow for the electrically adjustable color saturation with no polarizers by the addition of the dichroic dye; and (4) the comparison of theoretical and experimental electro-optical response curves aids to determine the shape of the droplets in a non-destructive way. Notably, the design of the study implies not just demonstration of these phenomena, but also the description of its physical background, which helps to further enhance the results for possible applications.

Thus we believe that our manuscript fits the selected special issue of Polymers journal, which plays an important role in journal mission to highlight the important advances of fundamentals and applications in polymer science.

Below is our point-to-point response to the specific questions. We hope the revised manuscript can be accepted for publishing in Polymers journal.

Here are some additional comments which might help the authors to improve their work:

In my opinion, the strong point of the work is the tunability of optical properties (specifically, colors) of the PDLC films, as it is highlighted in the abstract, however, the manuscript is mainly about calculations. I would very much recommend the authors reconsider the last section of the article and provide more experimental results proving the “tunability of color”. Now, it can hardly be said from Figure 6 that any color tunability takes place. Additional spectral characterization is also needed.

Answer:

Thank you for the comment. We agree that the tunability of optical properties is an important feature of these materials. In this manuscript, we focused on proof of this concept, thus we demonstrated our findings and described in detail its fundamental background. We keep in mind that the stronger effect can be delivered after the optimization of the material for this aim. In the current work, we have not intended to maximize the effect, but to show its principles, which is essential for further development. The difference in optical appearance depends on film thickness, type and concentration of dichroic dye, etc. At the moment we are unable to cover such a large task. At the same time, we have shown that the PDLC material with toroidal configuration of LC droplets can deliver this effect, and we are glad the Reviewer liked the idea.

We also noted that the term ‘color changes’ may mislead the reader, as one can understand it as significant hue changes (for example, red to green). In fact, the application of the electric field leads primarily to the changes in color saturation and brightness, as it is seen on the Fig. S2. Thus we corrected the use of the term ‘color’ across the manuscript. To follow the request of the Reviewer to demonstrate it in a more complete way, we added macro photographs with the area of several sq. millimeters of the sample with dichroic dye with no polarizers, which is close to the ‘naked eye’ observations of the film to Supplementary Information.

Comment:

The novelty of the work should be expressed clearly.

Answer:

Thank you for the comment. As is written in the first answer, the novelty of the work consists of the following: (1) we found that with cross-polarizers it is possible control the light transmittance by electric field in PDLC materials under study; (2) we examined the fundamental origins of the optical changes grant the unified value of reduced transition electric field for the droplets of different sizes and shapes; (3) we proven the concept that the PDLC with toroidal configuration of LC droplets allows for the electrically adjustable color saturation with no polarizers by the addition of the dichroic dye; and (4) we demonstrated that the comparison of theoretical and experimental electro-optical response curves aids to determine the shape of the droplets in a non-destructive way.

According to the Reviewers’ comment, we updated the abstract and conclusions sections of the manuscript.

Comment:

Figure 3 or Figure 4 seems unnecessary. There is no need to show quite similar results for different N. One of them can be easily shifted to supporting information, and the discussion on simulations could also be sharpened.

Answer:

Thank you for this valuable comment. We agree that one of these Figs. is unnecessary. Thus we did the following:

- we replaced figure for N0=2.7 with the plots of a/d(delta) at e=0 and e=4.2 (where the transition happens);

- we added data for N0=2.7 to the Supplementary;

- we added one curve for N0=2.7 to the Fig. 3 (previously Fig. 4, N0=4.3).

- we changed the text of the manuscript and SI accordingly.

Comment:

Few comments regarding the experimental part:

1) It is not indicated how much dichroic dye was used in CLC. 

Answer:

Thank you for this comment, we missed it. We added dye concentration data in Section 2.1: “The dichroic dye concentration was 2 wt %.”

Comment:

2) How CLC was dispersed in PiBMA? It is not possible to do since the polymer is solid. So, please indicate how it was done (solvent, etc.). 

Answer:

We agree with the Reviewer, as PiBMA polymer is a thermoplastic with glass transition temperature Tg = 65 °C. Therefore, we utilized SIPS technology using butyl acetate as a solvent to disperse the CLC into the polymer. We have added the additional description in Section 2.1:

“The sandwich-like cells have been manufactured by combining technology the solvent-induced phase separation (SIPS) and thermal-induced phase separation (TIPS). [5]. The glass substrates coated with ITO have been used. At the first stage (SIPS), CLC was added to the 10% solution of PiBMA in butyl acetate. Weight ratio of cholesteric and polymer components was CLC:PiBMA = 60:40. The prepared homogeneous solution was poured on the glass substrate and dried for 24 hours.”

Comment:

3) How the PDLC films were fabricated?

Answer:

The electro-optical PDLC cell (PDLC film under study) was assembled by heating the PDLC film between two substrates to a temperature above the Tg temperature of the polymer and the subsequent cooling to room temperature. CLC dissolved in the polymer at heating. The inverse process (phase separation) was observed at cooling (commonly referred as TIPS technology). It allows the assembly of an electro-optical cell and the control of the droplet size in the sample by the cooling rate. We updated the manuscript with additional details in Section 2.1 to make it more clear to the reader:

“At the second stage (TIPS), PDLC film was covered with the second substrate and heated to 80 °C under pressure. Then it was cooled down to room temperature for 5 hours, resulting in the PDLC cell under study. PDLC film thickness H was specified by 20 µm diameter glass microspheres (Duke Scientific Corporation), which were added to the solution of PiBMA in butyl acetate at the first stage (SIPS). The droplet sizes d (apparent diameter of the droplet) were determined by the sample cooling rate and were found to be in the range from 6 to 35 µm for 14 µm helix pitch CLC and in the range from 3 to 15 µm for 4 µm helix pitch CLC.”

Round 2

Reviewer 2 Report

The authors have reasonably addressed the comments. The revised version of the manuscript has been considerably improved. So, taking into account that the manuscript has been submitted for the special issues of Polymers on PDLCs I can recommend it for publication.